# A Pencil-Lead Immunosensor for the Rapid Electrochemical Measurement of Anti-Diphtheria Toxin Antibodies

**DOI:** 10.3390/bios11120489

**Published:** 2021-11-30

**Authors:** Wilson A. Ameku, Vanessa N. Ataide, Eric T. Costa, Larissa R. Gomes, Paloma Napoleão-Pêgo, David William Provance, Thiago R. L. C. Paixão, Maiara O. Salles, Salvatore Giovanni De-Simone

**Affiliations:** 1Oswaldo Cruz Foundation (FIOCRUZ), Center for Technological Development in Health (CDTS)/National Institute of Science and Technology for Innovation in Neglected Populations Diseases (INCT-IDPN), Rio de Janeiro 21040-900, Brazil; akira.ameku@gmail.com (W.A.A.); larissa.gomes@cdts.fiocruz.br (L.R.G.); paloma.pego@cdts.fiocruz.br (P.N.-P.); bill.provance@cdts.fiocruz.br (D.W.P.J.); 2Electronic Languages and Electrochemical Sensors Laboratory, Department of Fundamental Chemistry, Institute of Chemistry, University of São Paulo, São Paulo 05508-000, Brazil; vanneiva@usp.br (V.N.A.); eric.costa@usp.br (E.T.C.); trlcp@iq.usp.br (T.R.L.C.P.); 3National Institute of Bioanalytical Science and Technology, Campinas 13084-971, Brazil; 4Institute of Chemistry, Federal University of Rio de Janeiro, Rio de Janeiro 21941-909, Brazil; maiara@iq.ufrj.br; 5Cellular and Molecular Department, Biology Institute, Federal Fluminense University, Niterói 24020-141, Brazil; 6Epidemiology and Molecular Systematics Laboratory, Oswaldo Cruz Institute, FIOCRUZ, Rio de Janeiro 21040-900, Brazil

**Keywords:** electrochemistry, biosensor, diphtheria, square wave voltammetry, B-cell epitope, point-of-care

## Abstract

Diphtheria is a vaccine-preventable disease, yet immunization can wane over time to non-protective levels. We have developed a low-cost, miniaturized electroanalytical biosensor to quantify anti-diphtheria toxin (DTx) immunoglobulin G (anti-DTx IgG) antibody to minimize the risk for localized outbreaks. Two epitopes specific to DTx and recognized by antibodies generated post-vaccination were selected to create a bi-epitope peptide, biEP, by synthesizing the epitopes in tandem. The biEP peptide was conjugated to the surface of a pencil-lead electrode (PLE) integrated into a portable electrode holder. Captured anti-DTx IgG was measured by square wave voltammetry from the generation of hydroquinone (HQ) from the resulting immunocomplex. The performance of the biEP reagent presented high selectivity and specificity for DTx. Under the optimized working conditions, a logarithmic calibration curve showed good linearity over the concentration range of 10^−5^–10^−1^ IU mL^−1^ and achieved a limit of detection of 5 × 10^−6^ IU mL^−1^. The final device proved suitable for interrogating the immunity level against DTx in actual serum samples. Results showed good agreement with those obtained from a commercial enzyme-linked immunosorbent assay. In addition, the flexibility for conjugating other capture molecules to PLEs suggests that this technology could be easily adapted to the diagnoses of other pathogens.

## 1. Introduction

The respiratory and cutaneous disease diphtheria (DIPH) is caused by toxins released from the bacteria *Corynebacterium diphtheriae* and *C. ulcerans* during pharynx infections, tonsils, or skin. In severe cases, a visible pseudomembrane can develop in the upper respiratory tract along with polyneuritis and myocarditis. If not treated, the clinical presentation of the disease can quickly worsen, with an overall fatality rate from 5 to 10% [1,2]. Fortunately, DIPH is a vaccine-preventable disease with the efficacy of the toxoid-based vaccine varying between 54–87%. For herd immunity, 80–85% of the population needs to be vaccinated [3]. A major issue is that the immunity induced by the vaccines can wane over time, and any drop in the protection levels in a population could allow for an opportunistic return of this transmissible disease leading to an outbreak [1,4]. Previous studies suggest that 49% of the French adult population presents an antibody titer below protective levels [4]. To maintain the antibody titer at a protective level, booster shots are required. The World Health Organization (WHO) recommends booster vaccinations at 10-year intervals to everyone who lives in low- or non-endemic areas to ensure life-long protection [4].

Access to a simple, rapid serological test to determine the titer of antitoxin antibodies could be highly relevant to disease control. Currently, the titer of neutralizing antibodies in the serum is determined by assays for toxin neutralization and toxin-binding inhibition along with tests for passive hemagglutination, immunoblotting, and enzyme-linked immunosorbent (ELISA) assays [5]. Despite their accuracy, these diagnostics are time-consuming, require a laboratory facility and skilled personnel. The dependence on these conditions to perform the analysis is challenging, especially in resource-limited settings or out of standard laboratories [6,7,8]. point-of-care (POC) technologies are urgently needed that provide a decentralized assay, fast response, and reliable results to determine anti-diphtheria toxin (anti-DTx) antibody titer.

Electrochemical sensors could meet this demand through their characteristics; simple, portable, sensitive, easy-to-use, miniaturize, and operatable with a portable instrument [6,9,10,11,12,13]. When combined with biological recognition elements (i.e., enzymes, nucleic acid, antibodies, among others), electrochemical transducer-based POC devices have been developed to detect glucose [14,15,16], neurotransmitters [14], infectious agents, or their antibodies [12,17,18,19,20,21], pharmaceutical compounds [22,23,24], biomarkers [25] and DNA [26]. Furthermore, electrochemical sensors associated with biomimetic materials (i.e., nanozymes, synzymes, and metal complexes) display good robustness, long-term activity, and minimal matrix interference [9]. Numerous materials such as carbon and silver inks [25,27], tin and gold-sputtered layers [24,26], gold leaf [28], gold nanoparticles suspension [29], and microwires [30] have been used to fabricate disposable devices. However, pencil-lead electrodes (PLEs) stand out for their high electrochemical performance combined with the presence of dangling carbon bonds, carboxyl, carbonyl functional groups, and sp^2^-hybridized carbon atoms on the basal plane and edge [22,31,32]. These surface moieties permit a variety of different means to conjugate biological components that can be broadly applied to develop electrochemical sensors in the fields of clinical diagnosis [33], forensic [34], and environmental science [35].

There are significant concerns for antibody recognition in serum when whole antigens are used in serological tests due to the presence of a variety of epitopes that can react with antibodies against pathogens [36]. Rapid, sensitive, and specific electrochemical immunosensors for infectious diseases have been successfully developed with synthetic linear peptides [19,37]. When the peptides represent epitopes, which are antibody binding sites in pathogen proteins are positioned on the molecule’s surface [36], improvements can be achieved in the sensitivity and selectivity of diagnostic assays along with the elimination of cross-reactivity. Beginning with selecting two particular and reactive epitopes identified in the diphtheria toxin (DTx), this study focused on developing a low-cost and accurate electrochemical device to determine the titer of anti-DTx IgG in serum. The epitopes were synthesized in tandem and conjugated to a PLE integrated into a miniaturized three-electrode holder containing reusable reference and auxiliary electrodes using Ag/AgCl and a bare PLE, respectively. Antibodies captured by the peptides were measured by an indirect immunoassay using a secondary antibody conjugated with alkaline phosphatase that in the presence of hydroquinone (HQ) diphosphate, generated HQ electroactive-molecule detectable by square wave voltammetry (SWV). After optimization, a logarithmic calibration curve with good linearity over a wide concentration range and a low detection limit was realized. The final setup was evaluated for its capacity to measure the immunity level against diphtheria by quantifying IgG in actual serum samples and comparing it to a commercial ELISA. The high correlation in the results suggests that the peptide-modified PLE is a promising platform to assist in vaccination control programs, and the flexibility of the technology can be applied to a wide array of diseases.

## 2. Materials and Methods

### 2.1. Patients Serum Samples

Blood samples were collected from diphtheria/tetanus/pertussis (DTP)-vaccinated volunteers with no evidence of acute infection or known history of whooping cough or DIPH [38].

### 2.2. Chemicals and Reagents

*N*-(3-Dimethylaminopropyl)-*N’*-ethyl carbodiimide hydrochloride (EDC), *N*-hydroxysuccinimide (NHS), bovine serum albumin (BSA), and 2(*N*-morpholino) ethanesulfonic acid (MES), tris(hydroxymethyl)aminomethane hydrochloride (Tris-HCl), NaH_2_PO_4_, Na_2_HPO_4_, MgCl_2_, and NaCl were purchased from Sigma-Merck (St Louis, MO, USA). The 0.5 mm graphite pencil lead refills (2H, H, HB, 2B, 3B, and 4B from Pentel (Tokyo, Japan) were purchased from a local stationery shop. Goat anti-human IgG conjugated with alkaline phosphatase (secondary IgG antibody, sec-IgG) was purchased from Thermo (Waltham, MA, USA). The manufacturer’s recommendation was restored in 1 mL of deionized water; its final concentration was 0.6 mg mL^−1^. A commercial ELISA kit for anti-DTx quantitative immunoassay and negative/positive standard serum samples (human serum; negative for anti-human immunodeficiency virus antibody, hepatitis B-virus surface antigen, anti-hepatitis C virus antibody) were purchased from Serion Diagnostics (Würzburg, Germany). Hydroquinone diphosphate (diPho-HQ) salt was purchased from Dropsens (Llanera, Spain). Fuming HCl was purchased from Merck (Kenilworth, NJ, USA). Deionized water with a resistivity >18.1 MΩ cm was obtained from a Nanopure Diamond system (Barnstead, Dubuque, IA, USA) and used to prepare all solutions.

### 2.3. Fabrication of the Electrochemical Immunosensor

Working PLEs were fabricated from pencil lead refills (4B; 60 mm × 0.5 mm rods). Approximately half of the lead was coated with a glaze (Metal and Wood, Sherwin-Williams, Sumaré, Brazil) that insulated the lateral surface of the rod. The unglazed portion was used for connecting electrical leads. The glaze on the end that would constitute the electrode’s working surface was removed mechanically, bypassing the rod vertically on paper (Figure 1). The exposed surface was oxidized through chronoamperometry by immersion in PBS (pH 7.4) under vigorous stirring and applying +2 V vs. Ag/AgCl(KCl 3 mol L^−1^) for 50 s using a CompactState portable potentiostat (Ivium Technologies B. V., Eindhoven, The Netherlands). This was followed by four cycles of cyclic voltammetry (CV) with a scan rate of 100 mV s^−1^ over a potential range +0.4 to −1.4 V. Auxiliary, and reference electrodes were bare PLE and Ag/AgCl (KCl 3 mol L^−1^), respectively. Next, the PLE tip was immersed in 10 µL of 0.4 mol L^−1^ EDC/0.1 mol L^−1^ NHS prepared in 0.1 mol L^−1^ MES and 0.5 mol L^−1^ NaCl solution (pH 6.0) for 30 min to activate the carboxyl groups [39]. The PLE tip was then dipped for 30 min into PBS (pH 7.4) with peptide. The mentioned peptide (biEP) containing two DTx specific epitopes (GSFVMENFSSGGVDIGF) was synthesized as a linked tandem with the insertion of two glycine residues by solid-phase chemistry as previously described [38], and it was described briefly in Appendix A. After rinsing three times in PBS, the PLE + biEP was blocked overnight in PBS with 0.1% (*w/w*) BSA at 4 °C. To normalize for surface variations, the electroactive area of each electrode was estimated by chronoamperometry using the applied potential of 0.35 V for 150 s and the Cottrell equation in a 5 mmol L^−1^ [Fe(CN)_6_]^4−^ the solution was prepared in 0.1 mol L^−1^ KCl [40]. The parameters employed in the equation were: the number of electrons (*n* = 1), Faraday constant (F = 96,485 C mol L^−1^), the concentration of [Fe(CN)_6_]^4−^ = 5 × 10^−6^ mol cm^−3^, and the average diffusion coefficient value of the [Fe(CN)_6_]^4−^ in 0.1 mol L^−1^ KCl = 5.38 × 10^−6^ cm^2^ s^−1^ [41]. The one-way ANOVA test was performed (*n* = 3) with significance defined as *p* < 0.05 to check whether the device-to-device surface variation during the optimization of fabrication or analytical parameters—e.g., type of PLE, biEP and sec-IgG concentrations, serum dilution ratio, and time of exposition—was significant, as done in studies found in the literature [42,43]. Then, a two-tailed Students *t*-test with a confidence level of 95% was performed for pairwise comparisons. [42,43].

### 2.4. Electrochemical Assay to Detect Antibodies Anti-Diphtheria Toxin

Anti-DTx IgG detection was based on an indirect immunoassay wherein anti-IgG secondary antibodies conjugated with alkaline phosphatase (AP) hydrolyzed diPho-HQ to hydroquinone (HQ) resulted in changes in electrical signals, as shown schematically in Figure 1A. Briefly, anti-DTx IgG was captured onto the sensitized working surface of the PLE by a 30 min incubation at 25 °C in a 10 µL solution of patient serum (1:1000 in PBS) or control antibody. After rinsing in PBS, PLEs were immersed 30 min into an anti-human IgG secondary antibody solution conjugated with AP at 25 °C. Next, the PLE was inserted into a custom electrode setup that also contained reference (Ag/AgCl (KCl 3 mol L^−1^) and auxiliary (bare PLE) electrodes (Appendix A in Appendix A), which is described in Appendix B. The setup permitted the insertion of all three electrodes into a solution of 0.1 M Tris-HCl and 20 mmol L^−1^ MgCl_2_ (pH 9.8) with 5 mM diPho-HQ. The conversion to HQ was measured by square wave voltammetry (SWV), which employed 10 mV, 6.3 Hz, 10 mV, −0.6 to 0.6 V (vs. Ag/AgCl (KCl 3 mol L^−1^) as the values for amplitude, frequency, step, and applied potential window, respectively. Each cycle required 19 s, and a stable measurement was observed after the fourth cycle (76 s total time) corrected by the working electrode electroactive area calculated previously.

### 2.5. Enzyme-Linked Immunosorbent Assay

The anti-DTx IgG in sera was quantified by the indirect immunoassay using a commercial ELISA kit (Serion Diagnostics, Würzburg, Germany) following the manufacturer’s instructions and solutions. Briefly, each well of a 96-well ELISA plate was sensitized with diphtheria toxin antigen followed by washing and blocking with BSA. For the assay, 100 µL of a 1:100 dilution of patient serum was added along with negative and standardized positive controls provided with the kit. After a 60 min incubation at 37 °C, wells were rinsed with PBS and then incubated with anti-human IgG secondary conjugated with alkaline phosphatase (1:15,000) for 30 min at 37 °C. Next, the wells were rinsed four times was PBS followed by the addition of p-nitrophenylphosphatase and a 30 min incubation at 37 °C in the dark. Finally, 100 µL of 0.1 N NaOH/40 mM EDTA was added as a stop solution. The optical density was measured at 405 nm wavelength using a spectrophotometer (Multiskan SkyHigh, Thermo Fisher Scientific, Waltham, MA, USA).

### 2.6. Analytical Curve and Analysis of Blood Serum Samples

To assess the analytical performance, a standard curve was generated from the positive sample by diluting 0.35 IU mL^−1^ of positive IgG from the commercial ELISA kit into a negative patient serum to prepare a range of IgG concentrations from 10^−5^ to 10^−1^ IU mL^−1^. The indirect immunoassay was accomplished to detect IgG as described. The sera were diluted at a ratio of 1:1000 with PBS and used 1:50,000 diluted sec-IgG solution. In both cases, the incubation time was 30 min. They were analyzed by indirect immunoassay. The intensity of current density is proportional to the IgG concentration, correlated by the analytical curve.

## 3. Results

### 3.1. Preparation of a PLE Electrochemical Immunosensor

An electrochemical immunosensor was developed using graphite rods in pencil lead refills bound with a peptide consisting of two epitopes of DTx linked in tandem by two glycines. We previously identified the epitopes selected through a SPOT synthesis analysis [38]. As shown in Figure 1A, this arrangement would allow the stepwise attachment of anti-DTx IgG antibodies and AP-conjugated secondary IgG antibodies that would permit the generation of HQ, a redox molecule measurable by square wave voltammetry (SWV). Each fabricated PLE was intended to be a single-use sensor.

One-half of a PLE was coated with a glaze and physically manipulated on paper to expose the working area to create an electrically isolated surface (Figure 1B). Next, it was oxidized by applying a constant potential to create graphene oxide-like structures (Appendix A), which carry oxygenated groups such as carboxyl and aldehyde [44]. These were essential chemical groups for activation with EDC/NHS that would allow the covalent attachment of biEP through an amide bond [39]. To improve the ratio between the faradaic and non-faradaic currents, and before treatment with EDC/NHS, the PLE was reduced by cyclic voltammetry (Appendix A). The electrochemical treatment resulted in an electrode that presented a less bright and rougher appearance than before electrochemical treatment, as shown in the optical micrographs (Figure 1B), which improved device performance.

The choice of PLE type was critical for obtaining the desired electrochemical responses. In the market, pencil lead refills vary over a scale from 9H (harder/lighter) to 9B (blacker/softer). The H types have a high amount of wax and clay in their composition. In contrast, B types have a high content of graphite powder [45]. The pencil lead production process involves mixing graphite and clay particles (binding agent) and adding wax, resin, or polymers. As a final product, the intercalation of graphite particles in a clay matrix is obtained. The greater the softness of the pencil lead, the more graphite particles there will be in this matrix and, therefore, the more conductive this surface will be [46]. However, the graphite surface presents heterogeneous characteristics in structural terms, with basal plane regions (a region composed by conjugated sp^2^ carbon atoms) and edges (a structural region that presents defects, incomplete connections, sp^3^ sites, and the presence of functional groups, such as hydroxy, carboxyl, carbonyl) [47,48]. Consequently, these regions have different electrochemical reactivity, being the most the edge region [49]. Although this is a general understanding in the literature, no systematic studies have addressed the exact relationship between graphite amount and the electrochemical reactivity considering redox probes with different oxidation-reduction mechanisms. Moreover, the functional groups present on the graphite surface play an important role in the electrochemical reactivity of this surface since they are active sites [31]. In addition, oxygenated groups can be strategic points that help immobilize biorecognition molecules [50,51]. Here, the 4B exhibited a higher SWV average readout in response to the production of HQ compared to the others (Figure 2A). Those differences were significant (*p* < 0.01), except for HB (*p* = 0.17) and 3B (*p* = 0.28). The 4B pencil lead was chosen because its ratio for the signal to standard deviation was the highest (Figure 2A inset) and presented the most minor signal variation (0.3 ± 0.015 µA, *n* = 3, relative standard deviation (RSD) = 5%).

Next, the optimal concentration of biEP to conjugate to 4B was determined over a range of concentrations combined with an assay using the positive control included with the commercial ELISA kit. The biEP concentration on the PLE surface affected the current measured from the immunoreaction (Figure 2B). The ratio between SWV signals obtained after incubation in positive and negative samples only increased after exceeding 10 µg mL^−1^ of peptide (Figure 2B inset). Increasing the biEP concentration to 50 µg mL^−1^ resulted in the highest signal and the most significant ratio between positive and negative samples suggesting that more anti-DTx antibodies were captured, and non-specific interactions decreased. Statistically, significant differences (*p* < 0.001) were observed in pairwise comparison among signal ratios obtained with 50 µg mL^−1^ of peptide and others. Interestingly, the highest biEP concentration tested, 200 µg mL^−1^, leading to a drop in the SWV signal. One possible reason was that the higher density of biEP on the electrode blocked the surface, hindering the HQ diffusion. Therefore, the most suitable biEP concentration was determined to be 50 µg mL^−1^ and was used to produce all subsequent PLEs.

### 3.2. Optimization of Experimental Parameters, Reproducibility, and Stability

To optimize the analytical signal, the level of dilution for patient serum and secondary antibodies was evaluated and the incubation times. To fix the patient dilution factor, a 1hr incubation time was chosen for both the primary and secondary antibodies, diluted 1:5000. As the dilution of the positive antibody control was increased under these conditions, there was an increase in the signal intensity up to a dilution factor of 1000, followed by a decrease at the highest dilution factor of 5000 (Figure 3A). The non-specific binding of antibodies to the surface of the biEP sensitized PLE showed decreasing signals with higher dilutions. However, the ratio between positive and background reached a maximum difference at a dilution of 1:1000 (Figure 3A inset), which suggested that the biEP/IgG-specific interaction was favored, and a higher dilution ratio impaired slightly the signal due to a reduction in the availability of IgG. Significant differences (*p* < 0.001) were observed when comparing sera diluted by factors of 100 and 500 and sera diluted 1:1000, however, sera diluted 1:5000 had a similar response (*p* = 0.29) to sera diluted 1:1000. Therefore, as the serum diluted to 1:1000 had a slightly higher signal ratio, it was chosen as adequate. When using the optimal serum dilution factor at different incubation times, it was observed that 30 min was sufficient to achieve the highest signal and that a more prolonged incubation was not necessary (Figure 3B). A significant difference (*p* = 0.002) was observed comparing it with the signal obtained at 15 min, while with signals at 45 min and 60 min, there were no significant differences (*p* > 0.23).

Another critical factor was the sec-IgG concentration that identifies the biEP/IgG immunocomplexes. While a higher secondary antibody concentration leads to a higher SWV response (Figure 3C), it generates background signals with the negative control. The background signal decreased to the lowest levels at a dilution of 1:50,000, which did not impact the signal from the positive control and provided the most prominent sign-to-noise ratio (Figure 3C insert). The signal ratio obtained by diluting sec-IgG to 1:50,000 had a significant difference (*p* < 0.001) compared with those obtained at a dilution factor of 5000, 10,000, and 30,000. Although longer times had a minor influence on SWV responses (*p* > 0.05), incubation of 30 min was selected for providing stable signals (7 ± 0.3 mA cm^−2^, *n* = 3, RSD = 4%), and for being sufficient to obtain slightly higher signals (Figure 3D).

The reproducibility of the PLEs was analyzed by evaluating electrodes prepared on different days using the same protocol. An RSD of 6% in the SWV HQ response was calculated from 5 other measurements of a 10^−4^ IU mL^−1^ IgG solution, which demonstrated the practical reproducibility of the system (Appendix A). To test stability, PLEs prepared on the same day were stored at 4 °C in PBS. After 4 days of storage, the signal obtained from a 10^−4^ IU mL^−1^ IgG solution showed an RSD of 7% (*n* = 3) and a slight decay (5%) compared to using the PLE on the same day as its preparation, statistically, that difference was not a significant (*p* = 0.17) (Appendix A). The SWV current decreased by 19% and this time presented a significant difference in the signals obtained (*p* = 0.004) after 28 days of storage and presented an RSD of 10% (*n* = 3) (Appendix A).

### 3.3. Biosensor Performance

The performance of the PLEs was evaluated by indirect immunoassay incubating them in 0.1 mol L^−1^ PBS (pH 7.4) with an increasing quantity of anti-DTx IgG from 0 to 10^−1^ IU mL^−1^. The current density varied proportionally to IgG concentration, reaching a saturation region after 10^−3^ IU mL^−1^ (Figure 4A,B). After the linearization, the logarithmic analytical curve varied in a wide range of 10^−5^–10^−1^ IU mL^−1^ and showed a sensitivity of 800 µA cm^−2^ decade^−1^ (Figure 4C). The limit of detection (LOD) and the limit of quantification was 5 × 10^−6^ and 1.5 × 10^−5^ IU mL^−1^, based on three and 10 times the standard deviation, respectively. Considering the WHO report on a seroepidemiological study [2], the antibody levels for protective immunogenicity were designated on the graph. Antibodies levels below 10^−5^ IU mL^−1^ are considered non-protective while between 10^−5^–10^−4^ IU mL^−1^ confers primary protection, and above 10^−4^ IU mL^−1^ provides complete protection against DIPH.

To simulate a real-world application, the PLE was used to quantify anti-DTx IgG in serum samples collected from DTP-vaccinated volunteers, compared to a commercial kit. The two-tailed *t*-test showed there were no statistically significant differences (*p* > 0.5) at a 95% confidence level between the results from the two methods (*n* = 2), and Figure 4D shows the excellent correlation between them. They exhibited an RSD and the accuracy average varied between 8–20% and 92–117%, respectively (Table 1). The 70–120% variations and RSD ≤ 20% are acceptable for an analytical method [52,53]. All volunteers analyzed presented antibody levels corresponding to full-protection antibody levels.

## 4. Discussion

The observation that the protection afforded by vaccines against diphtheria can wane over time drives the necessity for diagnostic methods to measure the level of neutralizing antibodies. Further, alternatives are needed to replace the dependence on lab-based assays such as ELISAs and neutralization assays that are time-consuming and expensive. Here, we proposed implementing an electrochemical based on an electrode consisting of a pencil lead refill comprised of graphite that can be sensitized with a peptide to capture anti-DTx antibodies. Each PLE unit can be manufactured for a low cost and, in combination with our easily manufactured housing (Appendix A), would be mobile and provide rapid on-site results.

Conceptually, the development of an electrochemical immunosensor consists of an electron-conducting solid surface where the molecule of interest (antibody) can be captured, and its presence can amplify [19] or suppress [54] electrochemical signal. Thus, a key element is a biological component that can be immobilized onto the surface. Here, the PLE was modified with a peptide representing two selected epitopes in DTx linked in tandem (biEP) by two glycines, which improved the availability of the antigen to antibody [28,38]. This biEP works as a binding target for anti-DTx IgG and the epitopes are uniquely found in DTx, situated within a coiled structure on the protein surface that is available to the immune system and recognizable by B-cells antibodies [38]. Previous studies showed high specificity and sensitivity of 100% and 99.96%, respectively, in ELISA assays towards a panel of 92 sera with several diseases [38].

PLE modification with the biEP was verified by SWV recorded in [Fe(CN)_6_]^3−/4−^ solution. The size and insulating nature of the molecules of the electrode surface hindered the probe diffusion, which caused a decrease in the current as the peptide and blocking BSA were added (Appendix A). The prepared immunosensor would prove sufficient to provide the dynamic range needed to evaluate patient sera for neutralizing antibodies. The final results displayed a LOD was far lower than the toxin neutralization method, a sensitive and precise assay that detects antitoxins levels as low as 10^−3^ IU mL^−1^ [5]. Furthermore, the biEP displayed no cross-reactivity against serum from seropositive patients with Chagas, Chikungunya, Leishmaniosis, Pertussis, and COVID-19 disease (Appendix A).

The proposed device provided a reliable method for determining the titer of anti-DTx antibodies in serum and could be performed at room temperature and rapid measurement (76 s), compared with 30 min required for ELISA conducted at 37 °C. Furthermore, the simple construction, ease of electrode preparation and use, accuracy, and low cost suggest a high possibility for its use as a point of care diagnostic assay. Notably, the measurements can be performed in volumes <100 µL, which translates to the need for 0.1 µL of serum obtained from a finger prick sample of blood. Furthermore, considering that the measurements performed were defined by the biEP peptide, the surface of the electrode can be sensitized by peptides that represent other pathogens or diseases. Lastly, by converting to the spectroscopy impedance, a label-free imunossensor can be fabricated to eliminate the need for a secondary since the captured antibodies possess insulating characteristics.

## 5. Conclusions

A portable electroanalytical biosensor to assist in controlling diphtheria vaccination programs by accurately determining anti-DTx IgG titers in serum is described. A disposable working electrode was made from pencil lead pencil refills to create electrodes modified with a particular and reactive peptide consisting of two epitopes in tandem. This was integrated into a reusable and miniaturized electrode holder with reference (Ag/AgCl) and auxiliary (bare PLE) electrodes. The immobilized peptide on the electrode surface could capture anti-DTx IgG antibodies for measurement by an indirect immunoassay using an enzyme-conjugated secondary antibody that enzymatically hydrolyzed diPho-HQ into HQ, which was detected by square wave voltammetry. Under optimized working conditions, its logarithmic calibration curve exhibited good linearity across a wide concentration range of antibody concentrations that covered the protective levels of vaccinated individuals with a limit of detection far lower than the commonly used assays to determine the capacity for toxin neutralization. Notably, the results were in excellent agreement with those obtained from the commercial ELISA. Overall, our PLE setup could measure the immunity level against diphtheria toxin in serum samples, and the platform has the flexibility to meet the demands for other pathogens and their respective diseases.

## Figures and Tables

**Figure 1 biosensors-11-00489-f001:**
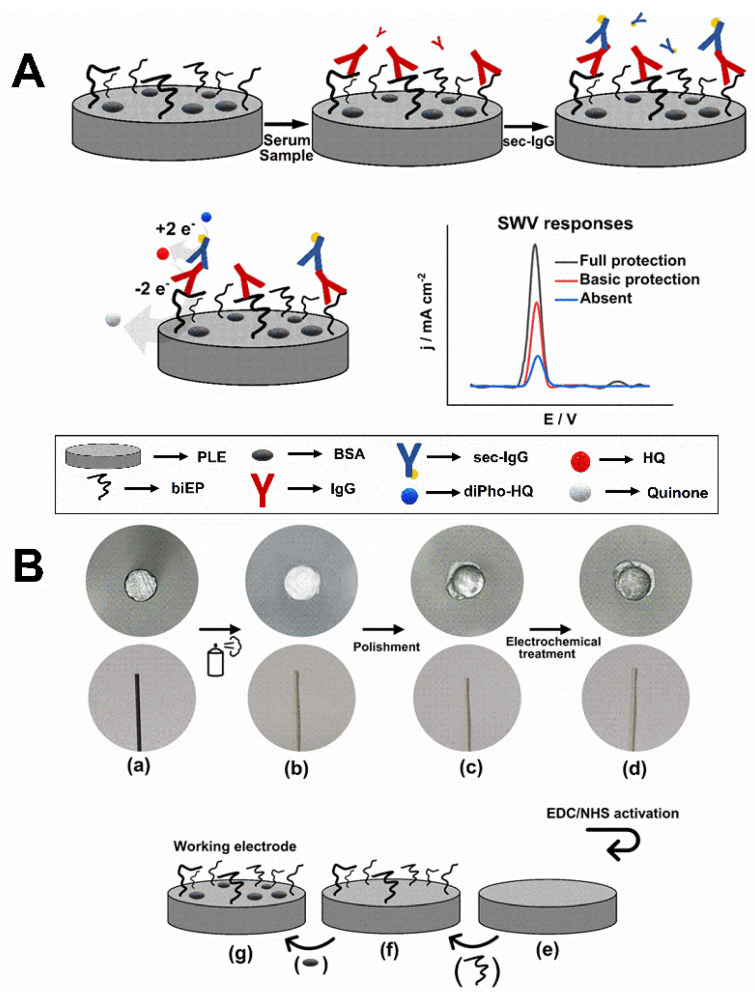
(**A**) Schematic representation of the indirect immunoassay to detect anti-DTx IgG in serum sample using PLE/biEP/BSA. The HQ SWV responses indicate the level of protection against diphtheria disease as full, basic, and absent. (**B**) Scheme of the process to fabricate the PLE. Optical micrographs of the side view and tip of PLE (Ø = 0.5 mm) in the following steps: (a) bare, (b) protected by glaze, (c) polished, (d) treated electrochemically followed by EDC/NHS activation. The drawings show the prepared surface (e), exposure to biEP (f), and followed by blocking with BSA (g).

**Figure 2 biosensors-11-00489-f002:**
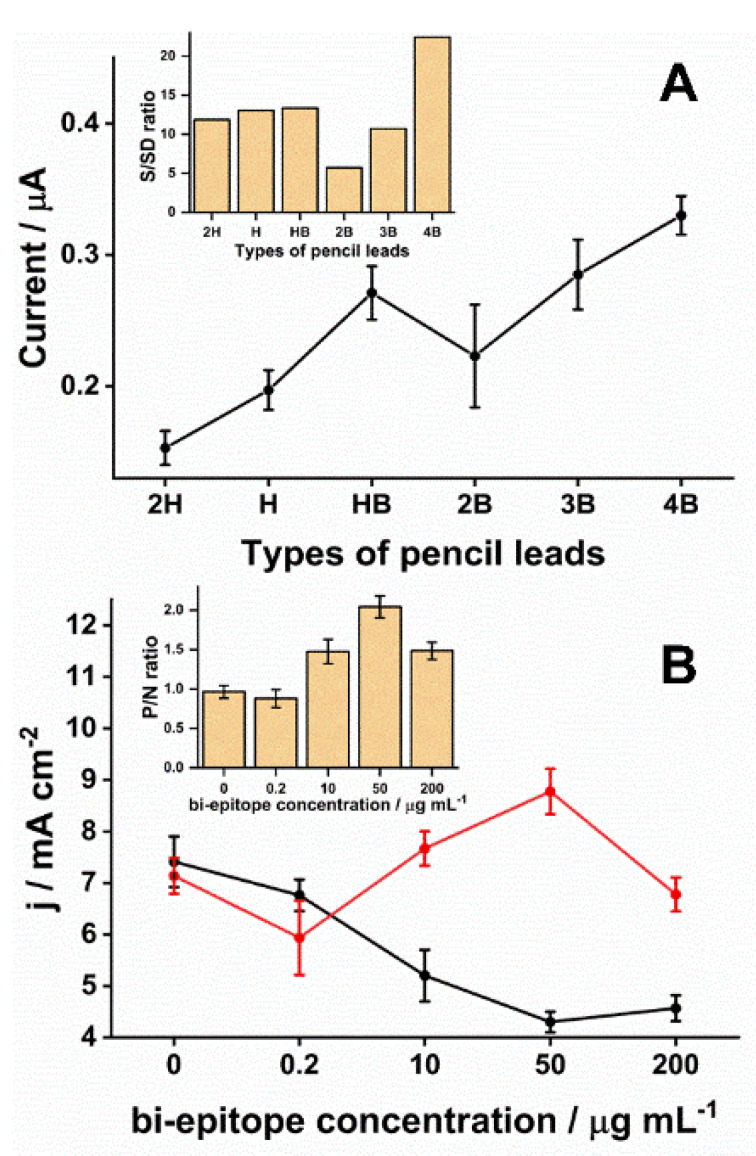
Optimization of the PLE fabrication. (**A**) As described in the Materials and Methods, six commercially available pencil lead refills (2H, H, HB, 2B, 3B, and 4B) were used to prepare electrodes. Each was incubated with a control anti-DTx antibody obtained from the commercial ELISA assay (0.2 IU mL^−1^) for 60 min. After rinsing, electrodes were incubated with anti-human IgG conjugated with alkaline phosphate diluted 1:5000 in PBS for 30 min. Lastly, electrodes were placed in a solution of 0.1 M Tris-HCl and 20mM MgCl_2_ (pH 9.8) with 5 mM diPho-HQ. Square wave voltammetry was performed with the parameters for amplitude, frequency, step, and applied potential window set at 10 mV, 6.3 Hz, 10 mV, −0.6 to 0.6 V vs. Ag/AgCl (KCl 3 mol L^−1^), respectively. Each PLE type was assayed with three independently prepared electrodes (*n* = 3). The inset graph represents the ratio between signal and standard deviation (S/SD) according to the kind of PLE. (**B**) The optimal concentration of the bi-epitope peptide to sensibilize the working surface of the 4B-PLE was determined by using a range of biEP concentrations from 0.2–200 µg mL^−1^. Next, PLEs were used to perform assays consisting of a 30 min incubation in either positive (0.2 IU mL^−1^ antibody solution from the commercial ELISA kit) or negative serum. After a 30 min incubation with an anti-human IgG antibody conjugated with alkaline phosphate (1:5000 in PBS), SWV was performed as described above. The average current densities from experiments performed in triplicate (*n* = 3) are plotted for positive (red) and negative (black) samples. The inset graph displays the ratio between positive and negative (P/N) measurements according to the biEP concentration.

**Figure 3 biosensors-11-00489-f003:**
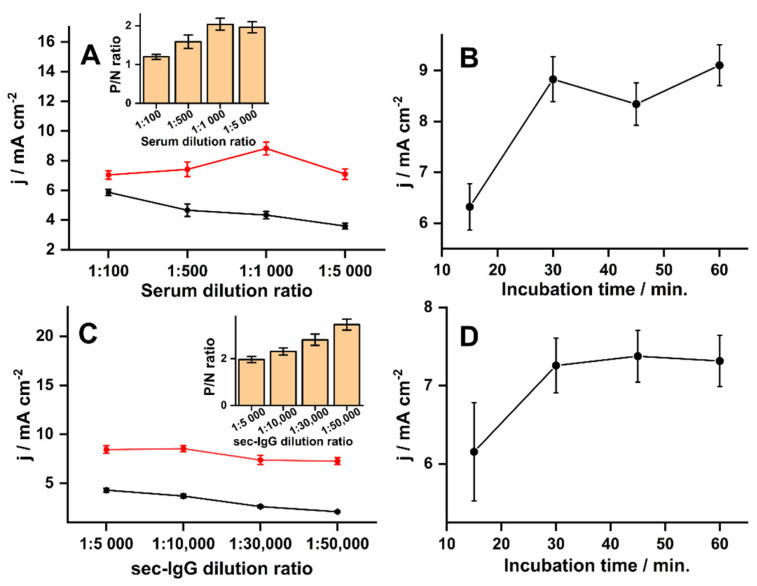
Variations in antibody dilutions and incubation times for optimized signals. All PLEs were prepared with 50 µg mL^−1^ biEP peptide and 4B refills with SWVs recorded in 0.1 mol L^−1^ Tris-HCl and 20 mmol L^−1^ MgCl_2_ (pH 9.8) with 5 mmol L^−1^ diPho-HQ. (**A**) Recordings after exposing PLEs for 30 min with a range of dilutions of positive (red) and negative (black) patient sera prepared in PBS, washing, and a 30 min incubation with secondary antibody (1:5000). (**B**) Recordings after exposing PLEs for different times with positive patient serum (1:1000 in PBS), washing, and a 30 min incubation with secondary (1:5000). (**C**) Recordings after exposing PLEs for 30 min to 1:1000 dilutions of positive (red) and negative (black) patient serum, washing, and 30 min incubations over a range of secondary antibody dilutions. (**D**) Recordings after exposing PLEs for 30 min to 1:1000 dilutions of positive (red) and negative (black) patient serum, washing, and incubations over a range of time with secondary antibody diluted 1:50,000. Error bars represent analysis in triplicate obtained with different electrodes (*n* = 3). The parameters of SWVs such as amplitude, frequency, step, and applied potential window were 10 mV, 6.3 Hz, 10 mV, −0.6–0.6 V (vs. Ag/AgCl (KCl 3 mol L^−1^), respectively. Inset graphs represent the ratio between positive and negative signals (P/N).

**Figure 4 biosensors-11-00489-f004:**
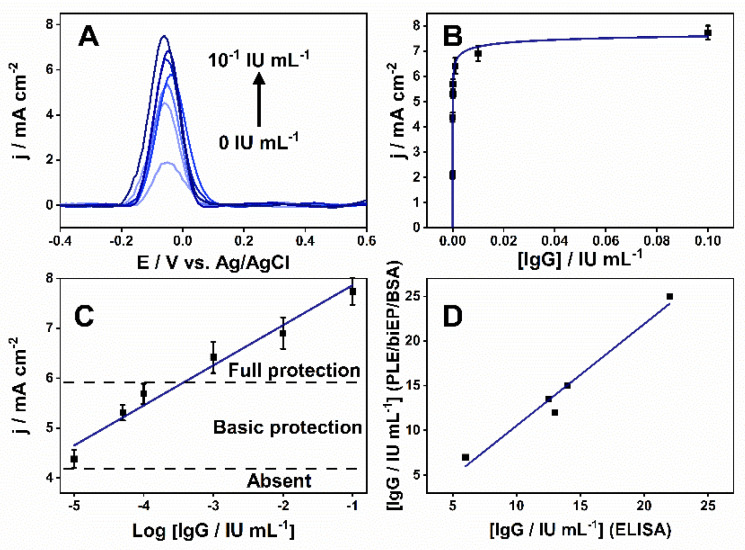
Detection of anti-DTx antibodies by biEP sensitized PLEs. (**A**) SWV measurements for different anti-DTx IgG concentrations (0, 10^−5^, 5 × 10^−5^, 10^−4^, 10^−3^, 10^−2^, 10^−1^ IU mL^−1^). (**B**) Relationship between the measured SWV current density and IgG concentration (0–10^−1^ IU mL^−1^). (**C**) The logarithmic analytical curve of IgG concentration range (10^−5^–10^−1^ IU mL^−1^) and current density with the mean and standard deviation from three independent measurements (n =3). The level of protection against diphtheria disease is indicated as full (>10^−4^ IU mL^−1^), basic (10^−5^–10^−4^ IU mL^−1^), and absent (<10^−5^ IU mL^−1^). The calibration equation and correlation are j/mA cm^−2^ = 0.8 mA cm^−2^ decade^−1^·log[IgG/IU mL^−1^] + 8.7 and R^2^ = 0.97, respectively. (**D**) Correlation between the measured anti-DTx level in patient serum samples obtained using PLEs and a commercial ELISA assay. The linearity of the relationship was y = 1.1x − 0.9 (R^2^ = 0.97).

**Table 1 biosensors-11-00489-t001:** Results of diphtheria IgG concentration determined using PLE/biEP/BSA and ELISA.

Serum Sample	C_PLE/biEP/BSA_ (IU mL^−1^)	C_ELISA_ (IU mL^−1^)	RSD (%)	Accuracy (%)
1	14 ± 2	13 ± 2	15	108
2	15 ± 3	14 ± 1	20	107
3	7 ± 1	6 ± 1	14	117
4	25 ± 4	22 ± 3	16	114
5	12 ± 1	13 ± 1	8	92

Data are presented as average ± standard deviation (*n* = 2).

## Data Availability

Data presented in this study are available on request from the corresponding author.

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
