# Peer review of "A Pencil-Lead Immunosensor for the Rapid Electrochemical Measurement of Anti-Diphtheria Toxin Antibodies"

_biosensors, 2021, doi:10.3390/bios11120489_

Round 1
Reviewer 1 Report
The comments are:
1- In figure 1, the scheme can be represented in better resolution.
2- On page 4, line 152, authors should point out N-(3-dimethyl aminopropyl)-N’-ethyl carbodiimide hydrochloride (EDC) and its concentration for activating carboxyl functional groups.
3- Some details reported in the method section, such as solid-phase peptide synthesis can be transferred to the supplementary information, and related references are cited in the main manuscript.
4- I could not define the similarity rate of the whole manuscript.
5- It is recommended that the introduction section be revised and represented concisely.
6- The conclusion of this manuscript should be revised carefully.
7- For comparison, I recommend that the previous studies be listed in one table showing the properties of applied materials and techniques.
Author Response
1-In figure 1, the scheme can be represented in better resolution.
R: We appreciated the observation. We repaired the resolution of the images. The following image was added to the manuscript, and as well it is available in MDPI’s system in 1,200 dpi.
2-On page 4, line 152, authors should point out N-(3-dimethyl aminopropyl)-N’-ethyl carbodiimide hydrochloride (EDC) and its concentration for activating carboxyl functional groups.
R: We are grateful for the valuable observation. We added the missing information in the text.
3-Some details reported in the method section, such as solid-phase peptide synthesis can be transferred to the supplementary information, and related references are cited in the main manuscript.
R: We would like to thank the advice. The referred section was transferred to the Supplementary Information.
4-I could not define the similarity rate of the whole manuscript.
R: We appreciated the observation. ANOVA and T-tests were done as required, the results were added in the text.
5-It is recommended that the introduction section be revised and represented concisely.
R: We are grateful for the valuable observation. The introduction presents enough information to understand the work. Therefore we did not modify it.
6-The conclusion of this manuscript should be revised carefully.
R: We would like to apologize for the unnecessary comment. We acknowledge and appreciate the reviewers' and editors' observations for the manuscript’s quality improvement. The statement has been removed.
7-For comparison, I recommend that the previous studies be listed in one table showing the properties of applied materials and techniques.
R: We would like to thank for the recommendation. The following table brings together previous studies related to Diphtheria and electrochemical detection, as shown in the previous version of the manuscript. However, it was removed from the current manuscript version because it lacks information and the difficult to establish a relationship between different Targeting analytes and Units.

Reviewer 2 Report
The author has designed a pencil lead-based electrochemical sensor for anti-diphtheria toxin antibodies. Some questions need to be answered to complete this manuscript.
- The quality of Figure 1 is too bad. Notes in the figure are not readable.
- I could not see the supplementary figures.
- Please explain the red curve drop at 0.2 ug/mL in Figure 2.
- As a well-developed working electrode material, the author needs to add more deep thinking about the electrode. For example, wax and clay amounts determine the pencil's hardness, but how does this affect the distribution of graphite powder? It definitely affects the binding site of the bioreceptor, but what is the relationship between binding efficiency and wax/clay amount?
- Page 4, line 152. The author only mentioned NHS. Where is EDC? EDC is more important than NHS, usually adding more EDC than NHS.
- What is the filling solution of the Ag/AgCl reference electrode? Or is it a pseudo-reference electrode?
- Figure 2 only has one figure instead of two figures.
Author Response
1-The author has designed a pencil lead-based electrochemical sensor for anti-diphtheria toxin antibodies. Some questions need to be answered to complete this manuscript.
R: We would like to thank the Reviewer for the valuable report. The answers follow under each question. We highlighted in green the changes in the text.
2-The quality of Figure 1 is too bad. Notes in the figure are not readable.
R: We appreciated the observation. We repaired the resolution of the images. The following image was added to the manuscript, and as well it is available in MDPI’s system in 1,200 dpi.
3-I could not see the supplementary figures.
We have checked the uploaded supplementary file, and its content can be seen normally.
4-Please explain the red curve drop at 0.2 ug/mL in Figure 2.
R: At a low concentration of epitopes, the graphite surface could not be appropriately covered, resulting in an unstable response where a drop in the average response was observed.
5- As a well-developed working electrode material, the author needs to add more deep thinking about the electrode. For example, wax and clay amounts determine the pencil's hardness, but how does this affect the distribution of graphite powder? It definitely affects the binding site of the bioreceptor, but what is the relationship between binding efficiency and wax/clay amount?
We appreciate the Reviewer's comment. A more detailed discussion has been added to the text (Section 3.1), addressing these aspects of the pencil lead surface.
6-Page 4, line 152. The author only mentioned NHS. Where is EDC? EDC is more important than NHS, usually adding more EDC than NHS.
R: We are grateful for the valuable observation. We added the missing information in the text.
7-What is the filling solution of the Ag/AgCl reference electrode? Or is it a pseudo-reference electrode?
R: We would like to thank the Reviewer question. The filling solution of the Ag/AgCl reference electrode is 3 mol L-1 KCl. This information has been added to the text (section 2.3) and the figure's captions.
8-Figure 2 only has one figure instead of two figures.
R: We appreciate the observation. We added the missing figure. The following image was added to the manuscript, and as well it is available in MDPI’s system in 1,200 dpi.

Reviewer 3 Report
I recommend this article be published after the inclusion of a couple of corrections.
- The figure's representation is not a good quality for Figure 1. It is not clear and discernable what the figure is telling you. Would you mind making the figure more clearly? The authors need to present a schematic showing the number of electrons transferred during the conversion of Hydroquinone diphosphate to hydroquinone.
- Figure 2 inset is too tiny. Please make it a bit larger so the audience can visually understand.
- In figure 3, all the small insets must be made more prominent.
- In figure 4, the SWV graphs changes are not visible. Other figures are not prominent and clear. The Legends are too tiny.
- Except for Table 1, other figures and graphs which contain error bar do all these readings are for n=2? I think these are more statistically viable if at least n=3 or more. Please clearly mention that in figure captions. What is the confidence interval doing these experimental data follow a normal distribution? Why is there no ANOVA testing is performed to see sensor to sensor variation? Statistical data validation test results should be more clearly present. No T-test has been completed. Authors can follow https://doi.org/10.1021/acsapm.0c01120; https://doi.org/10.1038/s41598-020-69636-1; for how to perform statistical calculations and data validation.
- The authors have mentioned so much information that it is not helping to highlight the sensor's working principle. I suggest the authors provide one section separate in discussion with schematic saying the working principle of the sensor.
- Why do authors for some electrochemical tests use 3mM Ferro/ferricyanide (for figure S2) and test 5mM Ferro/ferricyanide (for figure S3) as mentioned in lines 413 and 420? I guess if the redox analyte concentration is different, then the diffusion would have variations. It will result in varying current peaks. Why have the authors not used the same redox concentration? How would they match up with one -o- one comparison? In the SWV experiments, the authors used a low scan rate, so the concentration variation of the redox analyte would significantly affect the current density.
- Figure S5 and Figure S6 seem to be redundant. Would the authors please make both figures more precise and more perceivable? Try to upload a high-quality figure.
- Figure S4 has an error bar in the SWV graph! Authors either need to present all the three SWV graphs, or the authors can show one graph and inset a plot with error bars having currents?
- The authors used the abbreviation GRA but never mentioned what it is? It would be great if the authors could say all the abbreviations separately after the conflict of interest. So other than searching through the extensive text, it would be easier for readers to go through at a glance for all used abbreviations.
- Line 19: Would that be anti-DTx in the brackets?
- The last paragraph of the introduction is a hypothesis. So Authors need to mention what experiments and characterization they performed to achieve the hypothesis. I guess saying relevant theories and principles briefly is better.
- Conclusion: Line 389: How do authors know this section is not mandatory? Would you mind removing these unnecessary lines and letting the reviewers and editors of the journal decide what is needed?
Author Response
Reviewer #3 (changes highlighted in blue in the revised text)
1-I recommend this article be published after the inclusion of a couple of corrections.
R: We are grateful to the Reviewer for the valuable report. The answers follow under each question.
2-The figure's representation is not a good quality for Figure 1. It is not clear and discernable what the figure is telling you. Would you mind making the figure more clearly? The authors need to present a schematic showing the number of electrons transferred during the conversion of Hydroquinone diphosphate to hydroquinone.
R: We appreciated the observation. We repaired the resolution od the images and added the requested information. The following image was added to the manuscript, and as well it is available in MDPI’s system in 1,200 dpi.
3-Figure 2 inset is too tiny. Please make it a bit larger so the audience can visually understand.
R: We are grateful for the valuable observation. We repaired the figure. The following image was added to the manuscript, and as well it is available in MDPI’s system in 1,200 dpi.
4-In figure 3, all the small insets must be made more prominent.
R: We would like to thank the Reviewer for the observation. We repaired the figure. The following image was added to the manuscript, and as well it is available in MDPI’s system in 1,200 dpi.
5-In figure 4, the SWV graphs changes are not visible. Other figures are not prominent and clear. The Legends are too tiny.
R: We appreciate the Reviewer’s observation. We repaired the figure. The following image was added to the manuscript, and as well it is available in MDPI’s system in 1,200 dpi.
6-Except for Table 1, other figures and graphs which contain error bar do all these readings are for n=2? I think these are more statistically viable if at least n=3 or more. Please clearly mention that in figure captions.
What is the confidence interval doing these experimental data follow a normal distribution? Why is there no ANOVA testing is performed to see sensor to sensor variation? Statistical data validation test results should be more clearly present. No T-test has been completed. Authors can follow https://doi.org/10. 1021/acsapm.0c01120;https://doi.org/10.1038/s41598-02069636 -1; for how to perform statistical calculations and data validation.
R: We are grateful for the Reviewer's inestimable questions. All experiments were done in triplicate, except for Table 1. It was mentioned in figure captions. ANOVA and T-tests were done as required; the results were added in the text.
7-The authors have mentioned so much information that it is not helping to highlight the sensor's working principle. I suggest the authors provide one section separate in discussion with schematic saying the working principle of the sensor.
R: We are grateful for the suggestion. The sensor’s working principle was described in section 2.4 (Electrochemical assay to detect antibodies anti-diphtheria toxin) and at the beginning of the Results section.
8-Why do authors for some electrochemical tests use 3mM Ferro/ferricyanide (for figure S2) and test 5mM Ferro/ferricyanide (for figure S3) as mentioned in lines 413 and 420? I guess if the redox analyte concentration is different, then the diffusion would have variations. It will result in varying current peaks. Why have the authors not used the same redox concentration? How would they match up with one -o- one comparison? In the SWV experiments, the authors used a low scan rate, so the concentration variation of the redox analyte would significantly affect the current density.
R: We would like to thank the Reviewer for the question. The same concentrations were not used because the objective of each result presented in figures S2 and S3 were different and are not comparative. In the first figure (S2), the aim was to characterize the graphite surface at each stage of the surface treatment and how the electrochemical response of the redox probe was impacted. In this case, the technique used was cyclic voltammetry, which is generally used to show the electrochemical behavior of the electroactive molecule under study. In Figure S3, each step of surface modification and the consequent blocking of the electroactive surface by biorecognition agents was measured indirectly through the decrease in the current signal from the oxidation of Fe2+ (present in the potassium ferrocyanide complex) to Fe3+ (potassium ferricyanide). Thus, the smaller the redox probe current signal, the more blocked the graphite surface was, proving that the modifying agents are immobilized on this surface.
9-Figure S5 and Figure S6 seem to be redundant. Would the authors please make both figures more precise and more perceivable? Try to upload a high-quality figure.
R: We appreciate the inestimable observation. We have replaced both mentioned figures with Figure S1. The following image was added to the manuscript’s Supplementary Materials.
10-Figure S4 has an error bar in the SWV graph! Authors either need to present all the three SWV graphs, or the authors can show one graph and inset a plot with error bars having currents?
R: We appreciate the Reviewer’s observation. We repaired the figure. The following image was added to the manuscript’s Supplementary Materials.
11-The authors used the abbreviation GRA but never mentioned what it is? It would be great if the authors could say all the abbreviations separately after the conflict of interest. So other than searching through the extensive text, it would be easier for readers to go through at a glance for all used abbreviations.
R: We appreciated the comment. The list of abbreviations was included in the text.
12-Line 19: Would that be anti-DTx in the brackets?
R: We appreciated the observation. We made the change in the text.
13-The last paragraph of the introduction is a hypothesis. So Authors need to mention what experiments and characterization they performed to achieve the hypothesis. I guess saying relevant theories and principles briefly is better.
R: We would like to thank for the helpful question. The whole protein possesses multiple epitopes; some can react with non-target antibody generating cross-reactivity, as mentioned. Identifying specific epitopes for a given antibody, the cross-reactivity can be eliminated, as shown in Fig. S6. An indirect immunoassay using labeled secondary antibodies was carried out for this assay and others in this work. The resulting probe molecule (hydroquinone) enzymatically generated was detected by SWV.
14-Conclusion: Line 389: How do authors know this section is not mandatory? Would you mind removing these unnecessary lines and letting the reviewers and editors of the journal decide what is needed?
R: We would like to apologize for the unnecessary comment. We acknowledge and appreciate the reviewers' and editors' observations for the manuscript’s quality improvement. The comment has been removed.

Round 2
Reviewer 2 Report
Now the paper is ready to go.
Author Response
Thank you, we appreciate the Reviewer`s valuable effort.
Reviewer 3 Report
I am satified with the rebuttal. I recommed it for publication.
Minor: if you mention about performing 'T' test and ANOVA test try to cite some relevant work on same tests.
Author Response
Reviewer #3 (changes highlighted in yellow in the revised text)
I am satified with the rebuttal. I recommed it for publication.
We appreciate the Reviewer`s valuable report. The answer follow under the question
1 - Minor: if you mention about performing 'T' test and ANOVA test try to cite some relevant work on same tests.
We are grateful to the Reviewer for the valuable advice. We have added some relevant works found in the literature.
